# Radiologic Findings of Single Accessory Splenic Infarction in a Patient with Accessory Spleens in the Abdominal Cavity: A Case Report

**DOI:** 10.3390/medicina59040807

**Published:** 2023-04-20

**Authors:** Nan Xu, Yingchen Xu, Qiang Zhu

**Affiliations:** 1Department of Ultrasonography, Beijing Tongren Hospital, Capital Medical University, Beijing 100730, China; xunan835@126.com; 2Department of General Surgery, Beijing Tongren Hospital, Capital Medical University, Beijing 100730, China; xuyingchen66@163.com

**Keywords:** accessory spleen, infarction, ultrasound, computed tomography, magnetic resonance imaging

## Abstract

The presence of multiple accessory spleens in the abdominal cavity is typically limited to two, with cases involving a higher number being exceedingly rare. Concurrently, accessory spleen infarction is remarkably uncommon, primarily resulting from torsion of the vascular pedicle. In this report, we present a case of a 19-year-old male who experienced infarction in one of four accessory spleens. Imaging diagnosis proved challenging, with the definitive diagnosis being made through postoperative pathology, revealing no torsion in the affected accessory spleen. Following surgery combined with anti-inflammatory and analgesic treatment, the patient exhibited an uneventful recovery. No complications were observed at the 3-month follow-up. This case indicates the challenge and difficulty of diagnosing accessory splenic infarction without torsion in imaging diagnosis. Employing a multimodality approach and diffusion-weighted imaging may aid in confirming the diagnosis.

## 1. Introduction

An accessory spleen is a congenital healthy organ comprising splenic tissue separated from the main body of the spleen. The size of the spleen varies from 0.5 cm to 2.8 cm in a maximal caliber, and only one accessory spleen is usually seen [1]. Accessory splenic infarction is rare, and is caused by torsion of the vascular pedicle. However, the diagnosis of accessory splenic infarction is very challenging, and it needs to be differentiated from other diseases that cause acute abdomen. In addition, there are potential complications, such as rupture, hemorrhage, abscess, and compression symptoms, associated with enlargement [2,3,4,5,6,7,8]. Although accessory spleens are predominantly asymptomatic, they can provoke symptoms such as abdominal pain or cause an acute abdomen. In current study, we present a case of multiple accessory spleens with idiopathic accessory splenic infarction in the abdominal cavity. 

## 2. Case Presentation

A 19-year-old male patient complained of sharp pain in the left upper abdomen after heavy eating and drinking at a dinner. The pain persisted for 3 days, and was relieved after administration of anti-inflammatory medication and analgesia. Computed tomography (CT) of the lesion was performed on the 3rd day after the episode in a local hospital. Initially, a mass adjacent to the stomach, pancreas, and spleen, measuring 4.5 cm × 3.0 cm × 6.7 cm and with a well-defined margin, homogeneous soft tissue density, and edematous changes in the surrounding fat tissue, was revealed in an unenhanced CT image (Figure 1a). After three more days, enhanced CT imaging showed a lesion with pronounced annular enhancement and an increased size of approximately 4.8 cm × 3.3 cm × 7.4 cm. Additionally, three accessory spleens with the same density as the main spleen were found in the splenic hilum and its lower pole (Figure 1b–d). 

The patient was admitted to our hospital 14 days later for further diagnosis and treatment. He had no history of operative treatment or trauma, and had no family history, including of atrial fibrillation or coagulopathy. His abdomen was soft and flat, and a physical examination at presentation did not reveal tenderness. The patient had no abdominal tenderness, rebound tenderness, mass, or palpable liver. Upon admission, his vital signs were as follows: blood pressure, 125/94 mmHg; heart rate, 82 beats per min; respiratory rate, 20 times per min; and body temperature, 36.8 °C. The initial laboratory tests were as follows: white blood cell count: 5174/mm^3^ (normal range: 4000–10,000); hemoglobin: 15.2 g/dL (14.0–18.0); platelet counts: 403 × 109 g/L (125–350 × 109); serum: AST 38 IU/L (15–40), ALT 35 IU/L (0–41); prothrombin time: 11.2 s (10.5–15); activated partial thromboplastin time: 33.2 s (22.7–31.8); D-dimer: 0.12 mg/mL (0–0.3); and CRP: 0.33 mg/dL (0.0–0.5). Tumor markers were within the normal range. On the 12th day after the onset, abdominal sonography revealed a large, oval-shaped, well-defined solid mass surrounded by hyperechoic mesenteric fat tissue in the upper left abdominal cavity. The entity demonstrated low inhomogeneous echogenicity compared to the spleen (Figure 2a). No blood flow in- or outside the lesion was observed using a color Doppler mode (Figure 2b). To obtain more information, magnetic resonance imaging (MRI) was performed on the same day. The lesion appeared as a mass showing as hypointense on T1-weighted imaging (T1WI) and as hyperintense on T2-weighted imaging (T2WI), and a scattered hyperintense strip was present on its edges, possibly indicating hemorrhage (Figure 3a,b). The enhancement MRI pattern was similar to that of CT, but the size was reduced slightly to about 4.6 cm × 3.1 cm × 7.0 cm (Figure 3c). The apparent diffusion coefficient (ADC) value was lower than that of the spleens, indicating an extremely high degree of diffusion limitation (Figure 3d). There were no obvious abnormalities in the pancreas, main spleen, or kidneys, and no ascites or enlarged lymph nodes were found in the abdominal cavity nor in the retroperitoneum by either ultrasound or MRI imaging.

This case was indicative of a neoplasm, according to the above indicators found in imaging examinations. Thus, a laparoscopic excision was undertaken. Within surgery, a mass fed by a single vascular pedicle connected with the splenic artery was found in the upper left abdominal cavity, and that pedicle of vessels was not twisted. Macroscopically, the mass had a complete capsule, a gray-red surface, and a dark red profile with a size of approximately 6.5 cm × 4.5 cm × 3 cm (Figure 4a). Microscopically, large patches of infarcted areas with peripheral fibrous proliferation and slight vasculature were observed in the specimen (Figure 4b–d). The following images illustrate the large patches of infarcted areas, peripheral fibrous proliferation, and slight vasculature as mentioned in our description. Although we did not obtain images of the immunohistochemical staining, the immuno-histochemical profile of the tissue showed the following results: CD34 (+), Bcl-2 (+), STAT6 (−), S-100 (−), SMA (+), CD3 (+), CD20 (+), CD31 (+), and PAS (−). These markers collectively provided crucial information regarding the nature of the accessory spleen infarction and contributed to a comprehensive understanding of the case. Collectively, these data from the immunoreactivity report sufficiently supported our conclusions and reinforced the overall presentation of our case report.

The patient exhibited a successful recovery following surgical treatment and was discharged on the fourth postoperative day. An abdominal ultrasound and a CT were performed 1 month postoperatively, and no obvious abnormality was found. He was followed up on for 3 months, during which he was well, with no complications.

## 3. Discussion

Accessory spleens are not uncommon in imaging examinations, and, in general, have little clinical significance. Most occur singly (80.4%). Accessory spleens are most commonly found in the splenic hilum (75–90% of cases) and the tail of the pancreas. Accessory spleens are the most commonly encountered forms of developmental variations of the spleen [9]. The incidence of accessory spleens at autopsy is 10–30% in the American population and 4.5–24.3% in the Asian population [10]. They are more commonly found in males than in females, and their prevalence might be higher with certain genetic conditions, such as Down syndrome. The frequency of occurrence decreases proportionately with an increase in the number of accessory spleens [2]. The size of an accessory spleen may range from a few millimeters to a few centimeters, and rarely exceeds 4 cm in diameter [3]. There is no difference in histology between accessory and main spleens, and both are involved in the immune response [4]. In contrast to accessory spleens, splenosis typically occurs as an autograft after trauma or surgery. Histologically, it differs from accessory spleens in the absence of a hilum, the presence of poor capsule formation, and blood supply to the implant site [5]. The vascularization of accessory spleens is derived from the main splenic artery or other branches, such as the left gastroepiploic artery, short gastric artery, or the pancreas. The blood supply varies depending on the anatomical location, and despite being congenital, accessory spleens can function as normal splenic tissue, contributing to normal immune and hematologic functions. They undergo the same pathological changes as the main spleen. In addition, when functional spleen tissue needs to be removed, it is imperative to take into consideration the presence of an accessory spleen and remove it accordingly.

Accessory splenic infarction is a rare occurrence, and is mostly seen in children and young adults. It is rare (3.8%) in the elderly population (range: 14 days–75 years) [6,7,8,11,12,13,14,15,16,17,18,19,20,21,22,23,24,25,26,27,28,29,30,31,32]. There are no specific symptoms, although they are usually accompanied by abdominal pain, vomiting, fever, and even acute abdomen. Some patients have a history of chronic recurrent abdominal pain, which might be due to intermittent torsion resulting in blood flow reduction [16]. To date, there have been a total of 25 reports of 26 patients in the English literature [6,7,8,11,12,13,14,15,16,17,18,19,20,21,22,23,24,25,26,27,28,29,30,31,32]. Of these, 24 cases with infarction occurred due to torsion of the vascular pedicle with unknown risk factors, and the pathogenesis was presumed to be a lack of firm ligament fixation, strenuous exercise, or trauma [6,7,8,11,12,13,14,15,16,17,18,19,20,21,22,23,24,25,26,27,28,29,30]. However, according to Ozeki et al., the size, location, length, and origin of the pedicle of the accessory spleen had no significant correlation with torsion [27]. Additionally, Azzi and coworkers reported a 38-year-old woman with portal hypertension who suffered from an accessory splenic infarction, the mechanism of which was thought to be venous congestion caused by portal hypertension, leading to arterial blood flow stagnation and then infarction instead of torsion of the vessel pedicle [31]. 

In our case, no blood [24] system disease, arteriosclerosis, cardiovascular disease, or coagulation dysfunction was found after admission, and there was no history of trauma or surgery that could have caused splenosis. The laparoscopy showed that the supplying artery originated from the branch of the splenic artery and was normal in appearance. Therefore, this accessory spleen was diagnosed with infarction, but the infarction was not due to vascular pedicle torsion. We speculated that there were two possible reasons for the infarction. The first was the lesion’s location in a limited space, surrounded by the gastric cavity, pancreas, and spleen, causing it to be repeatedly compressed by these structures and, thus, leading to transient ischemia and intermittent pain over the previous four years. The second reason was the marked expansion of the stomach cavity due to heavy eating and drinking, leading to severe compression of the accessory spleen and causing both its infarction and sharp abdominal pain.

However, a clear diagnosis of accessory splenic infarction was dependent on surgical pathology rather than diagnostic imaging prior to surgery, as in previously reported cases and in our patient. Visualization of a vascular pedicle originating in the splenic area was the most important element in the diagnosis of an accessory splenic infarction caused by torsion [16]. However, among the 18 previously reported cases, only 11 (61.1%) were confirmed in imaging after resection [11,12,14,16,17,18,19,20,21,22,23,24,25,26,27,28,29,30], and the display rate of the vascular pedicle was not satisfactory with CT (62.5%, 10/16), ultrasound (22.2%, 2/9), or MRI (0/2). The reason why the ultrasound and MRI appeared weaker could be the inability of ultrasound scanning to pressure the abdomen, blood vessels with low caliber, and low spatial resolution of the MRI. Moreover, only 43.3% of the accessory splenic blood supply arteries stemmed from the splenic artery, so the difficulty of diagnosis was increased [33].

The imaging features of a normal accessory spleen, such as echogenicity, blood flow appearance on color Doppler, CT attenuation, iodine concentrations, signal intensity, ADC value, and enhancement pattern in CT and MRI were considered characteristic [34,35,36,37,38]. An infarcted accessory spleen was often descripted as a mass with an intrinsic hypoechoic level, a clear margin, a complete capsule, an oval contour, an annular hyperechogenic change due to edema of the surrounding mesenteric fat tissue, and avascular nature in ultrasonographic evaluation. In addition, the use of contrast-enhanced ultrasound could help to clarify the degree of loss of vascularization in the infarcted accessory spleen, whereas a hyper-enhanced change was shown in a normal accessory spleen [29,39]. On the CT scans, a sharply defined hypodense lesion was visualized in an infarcted accessory spleen. The lesion grew enlarged and surrounding inflammatory changes developed over time [16]. After the injection of a contrast agent, peripheral enhancement could be observed due to perfusion by the capsular vessels, and the degree of internal enhancement was relative to the progress of the infarction [16]. If the infarction was incomplete, an enhanced region could be seen adjacent to the hilus [19]. In MRI, there were three such cases reported in the literature in addition to our case. An infarcted accessory spleen manifested as hypointense and hyperintense focus on T1WI and T2WI, and, interestingly, a thin peripheral hyperintense rim was found in all cases on T2WI. Seo and coworkers suggested that it was representative of fibrosis [9]. However, no fibrous tissue was found in pathological specimens of Pérez’s cases, and it was proposed that the hyperintense rim could be caused by methemoglobin degraded from hemoglobin during the evolution of infarcted areas or slow blood flow in the residual capsular or subcapsular vasculature [14]. In summary, a single imaging method is inadequate; however, multimodal imaging and dynamic observation can greatly increase diagnostic confidence.

In our case, fibrous tissue and slight vascular proliferation around the infarcted area were seen in microscopic specimens. The onset time and MRI examination time of these three cases were different, indicating that the change in the MRI signal was extremely sensitive to the period of infarction. The spleen is the organ with the most restricted diffusion in the upper abdomen [37]. The ADC value of the infarcted accessory spleen was lower than that of the normal spleen tissue in our case, indicating a greater degree of diffusion restriction, probably due to the coagulation necrosis of the infarcted dense tissue. Thus, the combination of diffusion-weighted imaging and conventional MRI could greatly improve the detection of accessory spleen infarctions. Unfortunately, because of the rarity of this disease and the possibility of early intervention with surgery, it is difficult to realistically understand the imaging appearance of accessory splenic infarction. However, some signs and symptoms may help to identify accessory splenic infarctions. For example, sudden, sharp, or localized abdominal pain; a low-grade fever or elevated body temperature; tenderness or guarding; and increased inflammatory markers can be symptoms. Collectively, a thorough assessment of the clinical features, radiologic findings, and relevant laboratory data is essential for the accurate diagnosis and appropriate management of this rare condition.

The question of whether to perform a splenectomy for an infarcted accessory spleen has not been answered satisfactorily thus far. Involution or atrophy are the natural advancements of an infarcted accessory spleen [21]. Early diagnosis and timely surgical intervention are important in order to preserve the spleen [40]. It is essential to address any modifiable risk factors and monitor the patient for possible symptoms, conducting regular follow-ups to detect early signs of complications. However, even using advanced imaging modalities, we were unable to detect an infarcted accessory spleen with confidence in its early stages; thus, some severe complications, such as rupture, infection, and subsequent compression, might have occurred. Therefore, laparotomy or laparoscopic resection is almost always the treatment of choice. Furthermore, conservative therapy has been proven successful. Scirè and coworkers reported that a 10-year-old child with accessory splenic infarction was treated by acesodyne and antiphlogosis, and was relieved of abdominal symptoms with gradual atrophy on ultrasound follow-up images [24].

In this case report, we have identified several limitations that may impact the interpretation of our findings. First and foremost, our inability to definitively demonstrate the presence of a thrombus in the vessel feeding the infarcted accessory spleen due to a lack of direct intraoperative observation or pathological assessment contributes to the uncertainty of the actual cause. Additionally, while no torsion of the vascular pedicle was observed intraoperatively, it is important to consider that transient torsion might have been fixed spontaneously under general anesthesia, thus complicating our ability to exclude this potential cause of ischemia. Furthermore, the absence of sufficient pathological evidence for both transient torsion and thrombus formation led us to rely on a speculative approach to determine the most likely etiology of accessory splenic infarction in our case. Despite these limitations, we have made efforts to provide a balanced analysis of the potential causes, drawing support from the relevant literature and ensuring transparency in our discussion of the presented evidence.

## 4. Conclusions

Accessory splenic infarction without torsion of the pedicle is extremely rare, and the imaging characteristics are complex due to their association with various periods of the infarction. However, these methods, such as ultrasound, CT, and MRI, can provide valuable insights into the disease and inform clinical decision making, thereby facilitating appropriate and timely intervention.

## Figures and Tables

**Figure 1 medicina-59-00807-f001:**
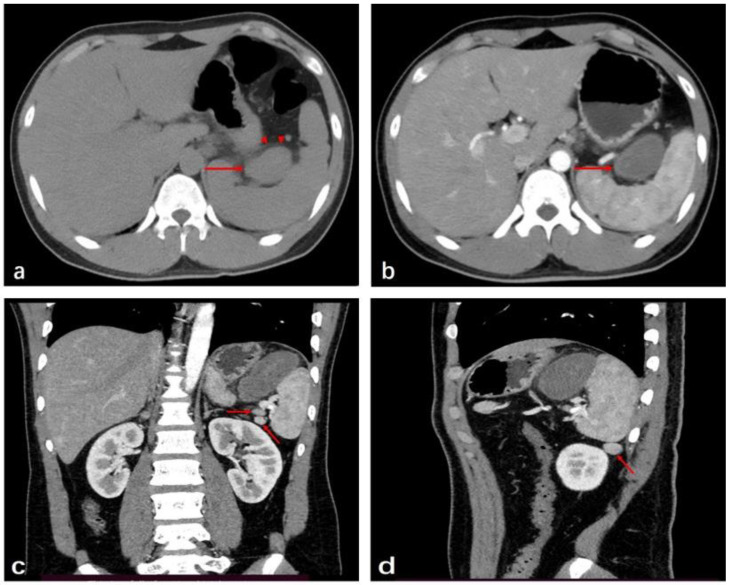
Computed tomography of the patient’s abdomen. (**a**): Non-contrast computed tomography revealed a mass-like soft-tissue density (red arrow) with adjacent inflammation (red arrowheads). (**b**): After enhancement, the mass was enhanced, and was circular (red arrow). (**c**): The CT enhancement showed two enhanced nodules at the hilum of the spleen (red arrow). (**d**): There was another enhanced nodule at the lower pole of the spleen (red arrow).

**Figure 2 medicina-59-00807-f002:**
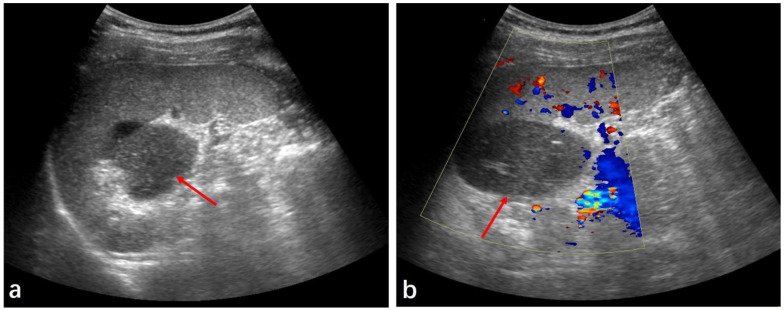
Ultrasonography of the patient’s abdomen. (**a**) Ultrasound showed a solid mass at the splenic hilum (red arrow). (**b**) Color Doppler showed no significant blood flow (red arrow).

**Figure 3 medicina-59-00807-f003:**
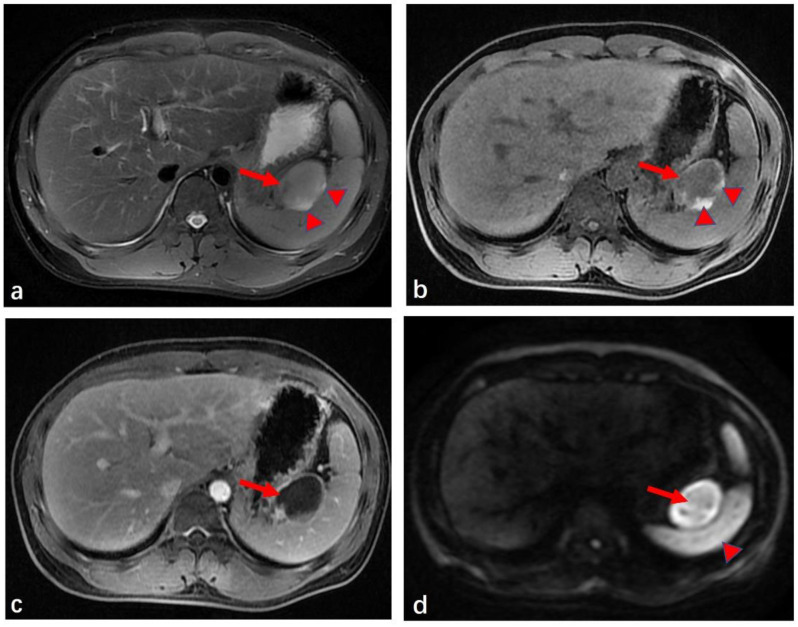
Magnetic resonance imaging of the patient’s abdomen. (**a**): T2WI shows a slightly hyperintense mass (red arrow) with some hyperintense areas (red arrowheads). (**b**): T1WI shows a homogeneous, hypointense mass (red arrow) with a thin peripheral hyperintense rim and some hyperintense areas (red arrowheads), similar to T2WI. (**c**): T1WI postgadolinium shows circular enhancement. (**d**): The DWI signal was significantly stronger (red arrow) than that of the spleen (red arrowhead).

**Figure 4 medicina-59-00807-f004:**
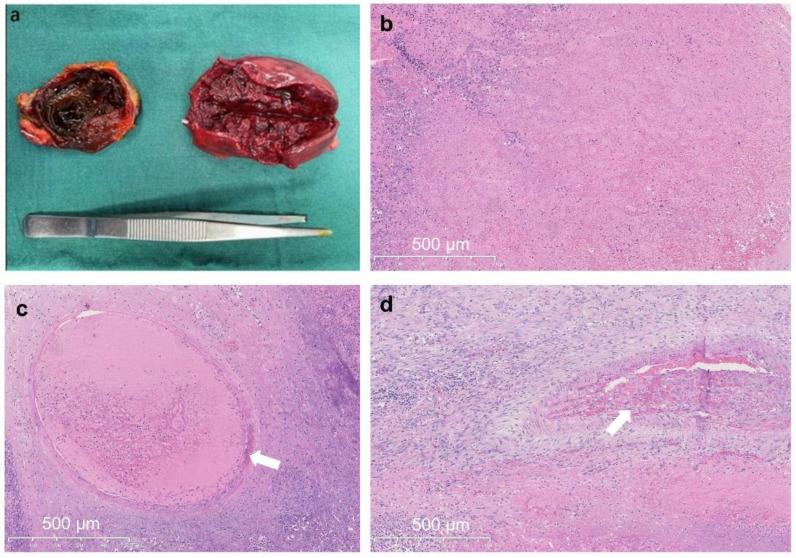
Accessory spleen histological evaluation. Image of the gross specimen (**a**) and pathological examination (HE × 100) (**b**–**d**). White arrows: thrombus formation (**c**) and thrombus organization (**d**). Scale bar: 500 µm.

## Data Availability

The authors confirm that the data supporting the findings of this study are available within the article.

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
