# Peer review of "Radiologic Findings of Single Accessory Splenic Infarction in a Patient with Accessory Spleens in the Abdominal Cavity: A Case Report"

_medicina, 2023, doi:10.3390/medicina59040807_

Round 1
Reviewer 1 Report
The Case Report by Nan Xu et al. is a well-documented case of accessory spleen infarction in a 19-years old male with four accessory spleens.
However, the authors should make some changes to improve the quality of their Case Report and to be accepted into Medicina
Major Comments:
1. Title. Please, change the title from “splenic infarction of multiple accessory spleens” to “accessory splenic infarction in a patient with multiple accessory spleens”
2. Discussion. Please, include more information regarding the congenital origin of the ectopic splenic tissue, data about the frequency in autopsy series, the localization of accessory spleens, and the vascularization of the ectopic tissue. Please include important references such as doi.org/10.4166/kjg.2021.071 and doi.org/10.1186/s12887-022-03484-y. Please include a short paragraph about the risk for new infarction in the other accessories spleens and if it is necessary to follow them.
Minor Comments:
3. Line 51. Please, initiate a new paragraph after the Figure legend
4. Line 54. Please, include more information regarding the physical examination (abdomen) and laboratory data (especially platelet count)
5. Line 123. The authors only show two possible reasons. One is about transient ischemia and the other is about the expansion of the stomach.
Author Response
Dear editor and reviewers,
We are grateful for the constructive feedback and valuable suggestions regarding our manuscript (medicina-2331916). The insightful comments from the reviewers have served as a strong foundation for enhancing the quality of our manuscript and its significance to the research community. In response to the reviewers' comments, we have made revisions to the manuscript and are submitting it for further consideration. All changes made to the manuscript are indicated in red, and a point-by-point response to the comments is provided in blue for clarity.
Reviewer #1
The Case Report by Nan Xu et al. is a well-documented case of accessory spleen infarction in a 19-years old male with four accessory spleens. However, the authors should make some changes to improve the quality of their Case Report and to be accepted into Medicina.
Response: We appreciate your positive feedback and are thrilled that you found our work to be likely to attract a wide readership. We have carefully considered your feedback and have made the according changes to the manuscript. We hope that these revisions have addressed your concerns and improved the overall quality of our manuscript.
Major Comments:
- Title. Please, change the title from “splenic infarction of multiple accessory spleens” to “accessory splenic infarction in a patient with multiple accessory spleens”
Response: Thank you for your valuable feedback on our manuscript. We appreciate your suggestion and agree that the proposed title more accurately reflects the content. The revised title of our manuscript now reads as below.
Title: Radiologic findings of single accessory splenic infarction in a patient with multiple accessory spleens in the abdominal cavity: a case report
- Discussion. Please, include more information regarding the congenital origin of the ectopic splenic tissue, data about the frequency in autopsy series, the localization of accessory spleens, and the vascularization of the ectopic tissue. Please include important references such as doi.org/10.4166/kjg.2021.071 and doi.org/10.1186/s12887-022-03484-y. Please include a short paragraph about the risk for new infarction in the other accessories spleens and if it is necessary to follow them.
Response: Thank you for the reviewer’s insightful comments and suggestions regarding the discussion section of our manuscript. Following your recommendations, we have revised the discussion section to include the requested information, as well as the suggested references. By incorporating this additional information and important references, we believe that our manuscript now presents a more comprehensive and informative discussion.
Page 6, line 131: Accessory spleens are most commonly found in the splenic hilum (75-90% of cases) and the tail of the pancreas. Accessory spleens are the most commonly encountered forms of developmental variations of the spleen [9]. The incidence of accessory spleens at autopsy was 10-30% in the American population and 4.5-24.3% in the Asian population [10]. They are more commonly found in males than in females, and their prevalence might be higher in certain genetic conditions, such as Down syndrome.
Page 7, line 145: The vascularization of accessory spleens is derived from the main splenic artery or other branches, such as the left gastroepiploic artery, short gastric artery, or the pancreas. The blood supply varies depending on the anatomical location, and despite being congenital, accessory spleens can function as normal splenic tis-sue, contributing to normal immune and hematologic functions.
Page 8, line 238: Early diagnosis and timely surgical intervention are important to preserve the spleen [40]. It is essential to address any modifiable risk factors and monitor the patient for possible symptoms, conducting regular follow-ups to detect early signs of complications.
Minor Comments:
- Line 51. Please, initiate a new paragraph after the Figure legend
Response: Thank you for the reviewer’s insightful comments, we have revised the manuscript according to your suggestion.
- Line 54. Please, include more information regarding the physical examination (abdomen) and laboratory data (especially platelet count)
Response: We appreciate your attention to detail and acknowledge the importance of providing a comprehensive clinical data. In response to your feedback, we have revised the manuscript to include a more detailed description of the abdominal examination findings, including the presence or absence of tenderness, palpable masses, and organomegaly. Additionally, we have expanded the laboratory data section, specifically focusing on the platelet count, as well as any other relevant hematological or biochemical parameters that may contribute to the patient's clinical presentation and outcome. We believe that these revisions will provide a more thorough understanding of the patient's condition and appreciate your guidance in enhancing the quality of our manuscript.
Page 2, line 65: He had no history of operative treatment or trauma, and had no family history, including atrial fibrillation or coagulopathy. His abdomen was soft and flat, and a physical examination at presentation did not reveal tenderness. The patient had no abdominal tenderness, rebound tenderness, mass, and no palpable liver. Upon admission, his vital signs were as follows: blood pressure, 125/94 mmHg; heart rate, 82 beats per min; respiratory rate, 20 times per min; body temperature, 36.8℃. The initial laboratory tests were as follows: white blood cell count 5,174/mm3 (normal range 4,000-10,000), hemoglobin 15.2 g/dL (14.0-18.0), platelet count 403 × 109 g/L (125-350 × 109), serum AST 38 IU/L (15-40), ALT 35 IU/L (0-41), prothrombin time 11.2 seconds (10.5-15), activated partial thromboplastin time 33.2 seconds (22,7-31.8), D-dimer 0.12 mg/ml (0-0.3), and CRP 0.33 mg/dL (0.0-0.5).
- Line 123. The authors only show two possible reasons. One is about transient ischemia and the other is about the expansion of the stomach.
Response: Thank you for pointing out the discrepancy in the number of possible reasons. In fact, there are only two reasons for the infarction of the accessory spleen in our manuscript. We apologize for any confusion caused due to the typo. we have ensured that the revised manuscript accurately presents and discusses the two possible reasons for the infarction.
Page 7, line 173: We speculated that there were two possible reasons for the infarction. first, the lesion's location in a limited space, surrounded by gastric cavity, pancreas, and spleen, which causes it to be repeatedly compressed by these structures, leading to transient ischemia and intermittent pain over the past four years; and next, the marked expansion of the stomach cavity due to heavy eating and drinking, which led to severe compression of the accessory spleen and caused its infarction and sharp abdominal pain.

Reviewer 2 Report
The authors reported in this manuscript a case of accessory splenic infarction in 19-year-old male. The infarcted accessory spleen was excised laparoscopically based on the suspicion of a neoplasm. My concerns are as follows.
1. The title “Radiologic findings of the splenic infarction of multiple accessory spleens in the abdominal cavity: a case report” may cause misunderstanding, because the infarction had not occurred in multiple accessory spleens, but just in one of four accessory spleens. The current title is better to be reconsidered to show clearly the point of this case report.
2. Considering the authors description described in Discussion “Unfortunately, because of the rarity of this disease and early intervention with surgery, it is difficult to realistically understand the imaging appearance of accessory splenic infarction.”, the definite diagnosis of this rare condition is hard to be made preoperatively like this case. Therefore, I would recommend the authors to focus on not just radiologic findings, but also the entire clinical features.
3. The last sentence in Introduction “Single imaging method is inadequate, howerver, multimodal imaging and dynamic observation can greatly increase diagnostic confidence.” should be mentioned in Discussion, not in Introduction, because this message might have been introduced from this case.
4. Although radiographic images might be core data, changes in laboratory data, especially that related to intravascular thrombus, such as D-dimer and thrombin-antithrombin complex(TAT), are better to be shown.
5. No red arrowheads indicating some hyperintense area are shown in Figure 3 a and b.
6. I wonder why and how the authors could have excluded the transient torsion of the vascular pedicle of the accessory spleen causing ischemia. They need to demonstrate the thrombus in the vessel feeding to the infarcted accessory spleen, because transient torsion might have been fixed spontaneously under general anesthesia.
7. Pathological photos are better to add, which could appropriately indicate “large patches of infarcted area with peripheral fibrous proliferation and small vasculature were observed in the specimen. Finally, the mass was confirmed as an accessory spleen infarction by immunohistochemical analyses.”.
8. Although the authors failed to obtain a correct preoperative diagnosis of accessory splenic infarction using ultrasound, CT, and MRI, I wonder how the authors could conclude “a multimodality approach using ultrasound, CT, and MRI may be useful in diagnosis of the disease”.
Author Response
Reviewer #2
The authors reported in this manuscript a case of accessory splenic infarction in 19-year-old male. The infarcted accessory spleen was excised laparoscopically based on the suspicion of a neoplasm. My concerns are as follows.
Response: We appreciate your positive comments. We have carefully considered your feedback and have made the according changes to the manuscript. We hope that these revisions have addressed your concerns and improved the overall quality of our manuscript.
- The title “Radiologic findings of the splenic infarction of multiple accessory spleens in the abdominal cavity: a case report” may cause misunderstanding, because the infarction had not occurred in multiple accessory spleens, but just in one of four accessory spleens. The current title is better to be reconsidered to show clearly the point of this case report.
Response: We appreciate your attention to detail and also agree that the current title may mislead readers into thinking that the infarction occurred in multiple accessory spleens instead of just one. In response to your feedback, we revised the title to more accurately reflect the content and focus of the case report. We suggest the following alternative title:
“Radiologic findings of single accessory splenic infarction in a patient with multiple accessory spleens in the abdominal cavity: a case report”
- Considering the authors description described in Discussion “Unfortunately, because of the rarity of this disease and early intervention with surgery, it is difficult to realistically understand the imaging appearance of accessory splenic infarction.”, the definite diagnosis of this rare condition is hard to be made preoperatively like this case. Therefore, I would recommend the authors to focus on not just radiologic findings, but also the entire clinical features.
Response: Thank you for the reviewer thoughtful suggestions and we agree that providing a more comprehensive view of the patient's clinical features, rather than just the radiologic findings, would offer a more complete understanding of this rare condition. we have added more clinical data and expanded our discussion to the clinical features, including the patient's history, presentation, physical examination, laboratory findings.
Page 2, line 65: He had no history of operative treatment or trauma, and had no family history, including atrial fibrillation or coagulopathy. His abdomen was soft and flat, and a physical examination at presentation did not reveal tenderness. The patient had no abdominal tenderness, rebound tenderness, mass, and no palpable liver. Upon admission, his vital signs were as follows: blood pressure, 125/94 mmHg; heart rate, 82 beats per min; respiratory rate, 20 times per min; body temperature, 36.8℃. The initial laboratory tests were as follows: white blood cell count 5,174/mm3 (normal range 4,000-10,000), hemoglobin 15.2 g/dL (14.0-18.0), platelet count 403 × 109 g/L (125-350 × 109), serum AST 38 IU/L (15-40), ALT 35 IU/L (0-41), prothrombin time 11.2 seconds (10.5-15), activated partial thromboplastin time 33.2 seconds (22,7-31.8), D-dimer 0.12 mg/ml (0-0.3), and CRP 0.33 mg/dL (0.0-0.5).
Page 8, line 230: However, some signa and symptoms may help to identify the accessory splenic infarction. For example, sudden, sharp, or localized abdominal pain, a low-grade fever or elevated body temperature, tenderness or guarding, and increased inflammatory markers. Collectively, a thorough assessment of clinical features, radiologic findings, and relevant laboratory data is essential for accurate diagnosis and appropriate management of this rare condition.
- The last sentence in Introduction “Single imaging method is inadequate, however, multimodal imaging and dynamic observation can greatly increase diagnostic confidence.” should be mentioned in Discussion, not in Introduction, because this message might have been introduced from this case.
Response: we appreciate with the reviewer’s valuable comment. we understand that moving this statement to the Discussion section will better situate it within the context of our case report findings.
Page 1, line 39: Although accessory spleens are predominantly asymptomatic, they can provoke symptoms, such as abdominal pain, or cause an acute abdomen. In current study, we present a case of multiple accessory spleens with idiopathic accessory splenic infarction in the abdominal cavity.
Page 8, line 218: Together, single imaging method is inadequate, however, multimodal imaging and dynamic observation can greatly increase diagnostic confidence.
- Although radiographic images might be core data, changes in laboratory data, especially that related to intravascular thrombus, such as D-dimer and thrombin-antithrombin complex (TAT), are better to be shown.
Response: Thank you for your valuable suggestion. We acknowledge the importance of presenting a comprehensive account of laboratory findings that may play a role in the diagnosis and understanding of the underlying pathophysiology of accessory splenic infarction. Therefore, we have added some laboratory data in the revised manuscript.
Page 3, line 70: The initial laboratory tests were as follows: white blood cell counts 5,174/mm3 (normal range 4,000-10,000), hemoglobin 15.2 g/dL (14.0-18.0), platelet count 403 × 109 g/L (125-350 × 109), serum AST 38 IU/L (15-40), ALT 35 IU/L (0-41), prothrombin time 11.2 seconds (10.5-15), activated partial thromboplastin time 33.2 seconds (22,7-31.8), D-dimer 0.12 mg/ml (0-0.3), and CRP 0.33 mg/dL (0.0-0.5).
- No red arrowheads indicating some hyperintense area are shown in Figure 3 a and b.
Response: We appreciate your attention to detail. The Fig 3 was revised to current state.
Figure 3. Magnetic resonance imaging of the patient’s abdomen. a: T2WI showed a slightly hyperintense mass (red arrow) with some hyperintense area (red arrowhead). b: T1WI showed a homogeneous hypointense mass (red arrow) with thin peripheral hyperintense rim and some hyperintense area (red arrowhead) similar to T2WI. c: T1WI postgadolinium shows circular enhancement. d: The DWI signal is significantly stronger (red arrow) than that of the spleen (red arrowhead).
- I wonder why and how the authors could have excluded the transient torsion of the vascular pedicle of the accessory spleen causing ischemia. They need to demonstrate the thrombus in the vessel feeding to the infarcted accessory spleen, because transient torsion might have been fixed spontaneously under general anesthesia.
Response: We appreciate your feedback. We understand the importance of verifying our conclusions and providing evidence to support our findings. In the Discussion section, we have included a thorough analysis of the potential causes of accessory splenic infarction, including transient torsion and thrombus formation. We have highlighted that both causes could play a role, but the actual cause remains uncertain due to the limitations in our assessment. In response to your comments, we have made the following changes to our manuscript:
Page 9, line 249: In this case report, we have identified several limitations that may impact the interpretation of our findings. First and foremost, our inability to definitively demonstrate the presence of thrombus in the vessel feeding the infarcted accessory spleen due to a lack of direct intraoperative observation or pathological assessment contributes to the uncertainty of the actual cause. Additionally, while no torsion of the vascular pedicle was observed intraoperatively, it is important to consider that transient torsion might have been fixed spontaneously under general anesthesia, thus complicating our ability to exclude this potential cause of ischemia. Furthermore, the absence of enough pathological evidence for both transient torsion and thrombus formation leads us to rely on a speculative approach to determine the most likely etiology of accessory splenic infarction in our case. Despite these limitations, we have made efforts to provide a balanced analysis of the potential causes, drawing support from relevant literature and ensuring transparency in our discussion of the presented evidence.
- Pathological photos are better to add, which could appropriately indicate “large patches of infarcted area with peripheral fibrous proliferation and small vasculature were observed in the specimen. Finally, the mass was confirmed as an accessory spleen infarction by immunohistochemical analyses.”.
Response: Thank you for your valuable suggestion and We agree that including these images would greatly enhance our presentation of the immunohistochemical analyses and the identified infarcted areas within the accessory spleen. In response to your comment, we have added representative high-quality pathological images. These images illustrate the large patches of infarcted area, peripheral fibrous proliferation, and small vasculature as mentioned in our description.
Figure 4. Accessory spleen histological evaluation. Image of the gross specimen (a) and Pathological examination (HE×100) (b-d). White arrow: thrombus formation (c) and thrombus organization (d).
Page 5, line 111: These images illustrate the large patches of infarcted area, peripheral fibrous proliferation, and small vasculature as mentioned in our description. Although we did not obtain images of the immunohistochemical staining, the immunohistochemical profile of the tissue showed the following results: CD34 (+), Bcl-2 (+), STAT6 (-), S-100 (-), SMA (+), CD3 (+), CD20 (+), CD31 (+), and PAS (-). These markers collectively provide crucial information regarding the nature of the accessory spleen infarction and contribute to a comprehensive understanding of the case. Collectively, these data from the immunoreactivity report sufficiently supports our conclusions and reinforces the overall presentation of our case report.
- Although the authors failed to obtain a correct preoperative diagnosis of accessory splenic infarction using ultrasound, CT, and MRI, I wonder how the authors could conclude “a multimodality approach using ultrasound, CT, and MRI may be useful in diagnosis of the disease”.
Response: The most crucial diagnostic method for splenic infarction remains pathological evidence. We acknowledge that despite our unsuccessful preoperative diagnosis in this case. However, for early diagnosis and evaluation, employing non-invasive imaging techniques is an ideal and reliable strategy. These methods, such as ultrasound, CT, and MRI, can provide valuable insights into the disease and inform clinical decision-making, thereby facilitating appropriate and timely intervention. Although imaging modalities may not always lead to a definitive diagnosis, their use in combination with other diagnostic information can aid clinicians in making an accurate assessment of the patient's condition. In response to your concern, we have revised our conclusion by emphasizing the potential benefits and limitations of the multimodality diagnostic approach (ultrasound, CT, and MRI) in assessing accessory splenic infarction.
Page 9, line 263: Accessory splenic infarction without torsion of the pedicle is extremely rare, and the imaging characteristics are complex due to their association with various period of the infarction. However, these methods, such as ultrasound, CT, and MRI, can provide valuable insights into the disease and inform clinical decision-making, thereby facilitating appropriate and timely intervention.
Once again, we greatly appreciate the time and effort the reviewers have put into evaluating our work and providing us with valuable feedback. We hope these changes have addressed reviewers’ concerns and provided a stronger support to understand the development of novel approaches to neural tissue engineering with the collagen materials and look forward to the opportunity to continue improving the manuscript.

Round 2
Reviewer 1 Report
The authors have carried out an accurate review and according to the reviewer's suggestions. The quality of his work has improved substantially. The current version can be published by Medicina
Reviewer 2 Report
I appreciate the authors' efforts to have revised this manuscript carefully based on my comments.